# Bladder Cancer Diagnosis and Follow-Up: The Current Status and Possible Role of Extracellular Vesicles

**DOI:** 10.3390/ijms20040821

**Published:** 2019-02-14

**Authors:** Eline Oeyen, Lucien Hoekx, Stefan De Wachter, Marcella Baldewijns, Filip Ameye, Inge Mertens

**Affiliations:** 1Flemish Institute for Technological Research (VITO), 2400 Mol, Belgium; eline.oeyen@vito.be; 2Centre for Proteomics (CFP), University of Antwerp, 2020 Antwerp, Belgium; 3Urology Department, Antwerp University Hospital (UZA), 2650 Edegem, Belgium; lucien.hoekx@uantwerpen.be (L.H.); stefan.dewachter@uantwerpen.be (S.D.W.); 4Pathological Anatomy Department, Antwerp University Hospital (UZA), 2650 Edegem, Belgium; marcella.baldewijns@uzleuven.be; 5Urology Department, General Hospital Maria Middelares Ghent, 9000 Ghent, Belgium; Filip.Ameye@AZMMSJ.BE

**Keywords:** extracellular vesicles, bladder cancer, biomarkers, urine

## Abstract

Diagnostic methods currently used for bladder cancer are cystoscopy and urine cytology. Cystoscopy is an invasive tool and has low sensitivity for carcinoma in situ. Urine cytology is non-invasive, is a low-cost method, and has a high specificity but low sensitivity for low-grade urothelial tumors. Despite the search for urinary biomarkers for the early and non-invasive detection of bladder cancer, no biomarkers are used at the present in daily clinical practice. Extracellular vesicles (EVs) have been recently studied as a promising source of biomarkers because of their role in intercellular communication and tumor progression. In this review, we give an overview of Food and Drug Administration (FDA)-approved urine tests to detect bladder cancer and why their use is not widespread in clinical practice. We also include non-FDA approved urinary biomarkers in this review. We describe the role of EVs in bladder cancer and their possible role as biomarkers for the diagnosis and follow-up of bladder cancer patients. We review recently discovered EV-derived biomarkers for the diagnosis of bladder cancer.

## 1. Current Diagnostic Tools for Bladder Cancer

Bladder cancer is the fourth most common cancer in men and the eighth most common cancer in women in the Western world [1]. Worldwide, it is the seventh most common cancer in men [2,3]. It occurs four times more in men than in women. 

Current diagnostic tools to detect bladder cancer are cystoscopy and cytology. Cystoscopy is an effective but invasive tool to detect bladder cancer tumors. Moreover, it has a low sensitivity for carcinoma in situ (Tis) and tumors can still be missed because effectiveness is operator-dependent, especially for the detection of recurrence [4]. Sensitivity and specificity range from 62 to 84% and 43 to 98%, respectively, depending on the type, stage, and grading of the tumor [5]. In addition, pain during urination (50%), urinary frequency (37%), visible hematuria (19%), and infection (3%) are experienced relatively frequently after flexible cystoscopy [6,7]. Urine cytology is a non-invasive diagnostic method used in clinical practice where voided or instrumented urine is examined for exfoliated cancer cells. The overall sensitivity ranges from 28 to 100%, with a median of 44%. It has a high sensitivity for high-grade tumors, but low sensitivity for low-grade tumors, ranging only from 4% to 31% [8,9,10]. Cytology is useful, particularly as an adjunct to cystoscopy, when a high-grade malignancy is present. A positive cytology indicates a urothelial tumor anywhere in the urinary tract. However, negative cytology does not exclude the presence of a tumor. Cytological interpretation is also user dependent and can be hampered, for example, by low cellular yield, urinary tract infections, and stones [11].

## 2. Urine-Based Biomarkers for Bladder Cancer

The use of urine-based biomarkers to detect bladder cancer seems to be an attractive alternative. Konety et al. (2006) defined the ideal bladder cancer biomarker as an objective, non-invasive, easily interpreted marker, possessing high sensitivity and specificity [12]. Urinary biomarkers are in direct contact with the bladder and can come in a variety of forms such as proteins, metabolites, DNA, different types of RNA, and single nucleotide polymorphisms (SNPs). Presence or variations in expression of those molecules could be linked to bladder cancer [13]. In the following section, Food and Drug Administration (FDA)-approved urine tests for bladder cancer are described. Table 1 gives an overview of these FDA-approved tests, the detected biomarkers and the assay type, as well as their sensitivity and specificity.

### 2.1. FDA-Approved Urine Tests

Nuclear matrix protein 22 (NMP22) can be detected in urine as a biomarker of urothelial cell death. This marker is frequently elevated in the urine of bladder cancer patients and can therefore be used in the detection of this disease. The NMP22® BC test kit and NMP22® BladderChek® are quantitative and qualitative enzyme immunoassay tests, respectively (originally Matritech Inc., Newton, MA, USA). The quantitative NMP22 BC test kit has a sensitivity of 69% and its specificity is 77% compared to a sensitivity of 58% and a specificity of 88% for the qualitative NMP22 test [14,15]. However, false-positive results are common since NMP22 is released from apoptotic cells also occurring during hematuria, inflammation, or infection [14,16,17,18].

The bladder tumor antigen (BTA) tests detect complement factor H-related protein, that is found in bladder cancer cell lines, using an immunoassay. The quantitative BTA (BTA TRAK®) test is performed in a specialized laboratory, whereas the qualitative BTA (BTA stat®) is a point-of-care test with an immediate result (Polymedco Inc., Cortlandt Manor, New York, USA). They have a sensitivity of 65% versus 64%, and a specificity of 74% versus 77%, respectively [12,14]. However, the specificity of both of these tests is significantly decreased since false positives have been noted to occur due to the presence of human complement factor H-related protein in blood. This is seen in various urological malignancies presenting hematuria such as urolithiasis, inflammation, recent instrumentation, other genitourinary malignancies, and intravesical Bacillus Calmette Guérin (BCG) therapy that causes local inflammation [12,14,19,20].

ImmunoCyt™/uCyt+™ is an immunocytochemical test that uses fluorescently labelled antibodies that are directed against three antigens: a glycosylated form of carcinoembryonic antigen and two mucins, specifically found on malignant exfoliated urothelial cells [21]. Mucins are high-molecular-weight glycoproteins, normally found on epithelial cell surfaces. In the case of bladder cancer, these glycoproteins are less glycosylated, thereby exposing a portion of the protein backbone. Sensitivity and specificity of this test are 78% and 78%, respectively [12,14,20]. The sensitivity of this test is higher than cytology, but the specificity is lower [22]. False positives are seen during infection or inflammation and there is a poor sensitivity in T2 bladder cancers. Moreover, interobserver variability exists; trained cytopathologists are therefore necessary [23]. It is only approved for the surveillance of bladder cancer patients [24].

UroVysion™ is a fluorescence in situ hybridization (FISH) probe set to detect bladder cancer (Abbott Molecular Inc., Des Plaines, IL, USA). It makes use of genetic markers in contrast to the previously mentioned tests, that were based on protein markers. It is FDA-approved for the diagnosis and surveillance of bladder cancer [24]. FISH is a technique that uses fluorescently labelled DNA probes to assess cells for genetic alterations [25]. Voided urine is analyzed for exfoliated urothelial cells that are hybridized on a slide. These are further examined for chromosomal aberrations observed in bladder cancer: aneuploidy of chromosomes 3, 7, and 17, and a loss of locus 9p21 [24,26]. In a meta-analysis, the sensitivity of the test was stated to be 72% and the specificity to be 83% in the context of an equivocal cytology [27]. Another recent meta-analysis of studies of UroVysion™ has calculated its sensitivity and specificity in detecting bladder cancer at 63% and 87%, respectively [14]. The lack of sensitivity for low-grade bladder cancers remains [28].

As depicted above, the sensitivity of most of these tests increases with higher tumor stage or grade [28]. In some tests, false positives are seen due to inflammation and hematuria, for example. This hampers the diagnosis of recurrences [14,19,23]. The FDA-approved tests have not replaced the current diagnostic standards of urine cytology and cystoscopy. In order to improve bladder cancer diagnosis, extensive research is being carried out in the search for sensitive and specific biomarkers. Different types of urine biomarkers can be used. In the following section, we give a non-exhaustive overview of potential urine biomarkers for bladder cancer diagnosis (Table 2).

### 2.2. Non-FDA Approved Urine Biomarkers

#### 2.2.1. Non-FDA Approved Urine Protein Biomarkers

Examples of possible urine protein biomarkers are two transcription factors: urothelial bladder carcinoma 1 (BLCA-1) and urothelial bladder carcinoma 4 (BLCA-4). These are NMPs, isolated from human bladder tumors. They show an elevated expression early in the development of bladder cancer and can potentially be used as biomarkers at an early stage, even before the appearance of a visible tumor [12]. BLCA-4 can be found in the early stage of bladder cancer, but is not expressed in normal tissues [30,31]. The enzyme-linked immunosorbent assay (ELISA) detecting the presence of BLCA-4 in urine has a sensitivity of 89 to 96% and a specificity of 90 to 100%, whereas the assay for urinary BLCA-1 shows a sensitivity of 80% and a specificity of 87% [32]. BLCA-1 and BLCA-4 are potential biomarkers for the diagnosis of bladder cancer in an early stage, but they still need further validation [24]. Hyaluronidase degrades hyaluronic acid into small fragments that promote angiogenesis. The expression of hyaluronidase correlates with invasive potential in bladder cancer. However, detecting low-grade tumors using this biomarker is difficult [33,34]. Intracellular proteins called cytokeratins are part of urothelial cells and are released in urine following cell death. Overexpression of certain cytokeratins is associated with bladder cancer. Fragments of cytokeratin 8 and 18 are detected in the urinary bladder cancer antigen (UBC)-ELISA and UBC-rapid tests. The tests have a low sensitivity: 21 to 84 % [23,35]. For the UBC-rapid test, Ecke et al. (2017) reported a sensitivity of 87% for detecting carcinoma in situ, 30% for low-grade non-muscle invasive bladder cancer (NMIBC), 71% for high-grade NMIBC and 60% for high-grade muscle invasive bladder cancer (MIBC) [36]. Another cytokeratin, cytokeratin 19 fragment called CYFRA 21-1, is detected with the CYFRA 21-1 test. This test has a high rate of false positives [37,38,39]. Survivin, an anti-apoptotic protein, is also elevated in bladder cancer [24,40,41,42]. It induces changes that are associated with tumor cell invasiveness. Survivin levels are associated with bladder cancer presence and higher tumor grade. However, further studies are necessary [43]. Chen et al. (2013) also found that epidermal growth factor (ProEGF) was significantly decreased in bladder cancer patients, whereas serum amyloid A4 (SAA4) was significantly increased, when comparing hernia and bladder cancer patients [44]. A combination of these two showed a higher diagnostic value in differentiating bladder cancer patients from controls. In addition, six apolipoproteins (APO), namely, APOA1, APOA2, APOB, APOC2, APOC3, and APOE, were present at elevated levels in the patient population. It is questionable whether apolipoproteins can be highly specific markers for bladder cancer, since these proteins can also be found in blood and hematuria is a non-specific symptom for bladder cancer. However, most studies described no effect of hematuria on these biomarker levels [45]. The upregulation of SAA4 was also seen in kidney cancer [44]. Other examples of potential protein markers are C-C motif chemokine 18 (CCL18), plasminogen activator inhibitor 1 (PAI-1), and cluster of differentiation molecule 44 (CD44) [46]. 

In addition to protein biomarkers, metabolites may also be of interest as urine cancer biomarkers [47,48]. For example, increased levels of lipids may reflect a higher tumor cell proliferation rate and increased lipid membrane remodeling. Wittmann et al. (2014) performed a metabolomic profiling of urine for bladder cancer biomarker discovery and defined metabolite biomarkers as palmitoyl sphingomyelin, lactate, phosphocholine, guanidinoacetate, branched chain amino acids (BCAAs), (iso)leucine and valine, adenosine, and succinate for bladder cancer [49].

#### 2.2.2. Non-FDA Approved Urine Genetic Biomarkers

In the search for genetic biomarkers to detect bladder cancer and monitor further treatment results, multiple aspects can be assessed such as DNA, cell-free DNA (cfDNA), and circulating RNAs: microRNA (miRNA), long non-coding RNA (lncRNA), messenger RNA (mRNA), and small interfering RNA (siRNA). As examples of genetic biomarkers for bladder cancer, mutations in the fibroblast growth factor receptor 3 (FGFR3) oncogene are frequent in low-grade non-muscle invasive bladder cancer (NMIBC) tumors. Mutations in RAS (Rat sarcoma) oncogenes occur in 13% of all bladder tumors [50,51,52]. The sensitivity for detecting bladder cancer by FGFR3 mutation analysis was 58% in a study undertaken by Zuiverloon et al. (2010) [52]. Mutations in p53 genes are seen more often in high-grade NMIBC [53]. This results in dysregulation of the RAS-MAPK (mitogen-activated protein kinase) pathway. Mutations in these genes are a strong indicator for bladder cancer. However, if no tumor cells are present in a bladder cancer sample, this results in a false-negative result. Cxbladder® is an example of a urine-based assay that is marketed but is non-FDA approved. It quantifies four mRNAs which are overexpressed in bladder cancer: cycline-dependent kinase 1 (CDK1), homeobox A13 (HOXA13), midkine (MDK), and insulin-like growth factor-binding protein 5 (IGFBP5). Chemokine receptor 2 (CXCR2) is used to reduce false-positive results due to inflammation [23]. It is reported that Cxbladder® has a sensitivity of 74% and a specificity of 82% [53]. In addition to this marketed test, other mRNAs are also described as potential urine biomarkers for bladder cancer. For example, paraoxonase-2 (PON2) mRNA levels in urinary exfoliated cells from bladder cancer patients can be used as a biomarker [54]. Moreover, microRNAs (miRNAs) have great potential as biomarkers [55,56,57,58,59]. De Long et al. (2015) concluded that increased numbers of miRNAs are detected in the urine of bladder cancer patients, also depending on the stage and grade [59]. 

Epigenetic factors also play an important role in the development of bladder cancer and can be also used as biomarkers. One of those factors are alterations in DNA methylation which may change the gene expression and can ultimately lead to the development and progression of urinary bladder cancer. This can potentially be used to diagnose urinary bladder cancer [60]. Several studies have revealed the role of methylated genes [60,61,62,63,64]. However, further validation of these markers is required and is still hampered by the expensive, time-consuming and highly specialized molecular genetic techniques needed to detect epigenetic alterations [13].

Here, we only give an overview of the different types of urine biomarkers that can be used for bladder cancer diagnosis and some examples. Since many biomarker discovery studies are carried out on bladder cancer diagnostics, many more potential biomarkers are described in the literature and can be found in recent reviews such as [13,66,67]. Although some of these biomarkers are very promising, they are currently not used in clinical practice for several reasons: no comparative research with a sufficient sample size has been conducted to validate these biomarkers as an adjunct to or a replacement of cystoscopy. They show overall a low sensitivity and therefore miss a significant portion of bladder cancer patients and at the same time may result in false-positive outcomes [14]. Furthermore, the detection of low-grade tumors is still limited and the accuracy for initial diagnosis is higher than in the case of a recurrence. Further research is needed to find better combinations of biomarkers to develop more sensitive and specific tests, especially to diagnose early-stage and low-grade tumors [14].

## 3. Extracellular Vesicles

### 3.1. Nomenclature

In recent years, there has been a growing interest in extracellular vesicles (EVs). EVs are membrane vesicles that are released by most cells into the surrounding extracellular environment and can be divided into different subgroups: apoptotic bodies, microvesicles, and exosomes. They all have different size ranges and biogenesis. Apoptotic bodies (50–5000 nm) are released by cells undergoing cell death by apoptosis. Microvesicles (50–1000 nm) are large membranous vesicles that are shed directly from the plasma membrane. Exosomes (40–100 nm) are small nano-sized vesicles originating in the late endosomal compartment by the inward budding of multivesicular bodies (MVBs). Exosomes are released from normal, diseased, and tumor cells into the extracellular environment by fusion of intracellular MVBs with the plasma membrane [68,69,70,71]. They are present in all body fluids, such as saliva [72], blood (plasma [73] and serum [74]), breast milk [75], and urine [76,77]. Their cargo consists of nucleic acids (mRNA, miRNA, etc.), proteins, and lipids [78,79]. This is a molecular fingerprint representing the cell of origin [80]. Previously, the terms exosome, microparticle, exosome particle, etc., were used for the isolated vesicles. However, there is still a lack of widely accepted specific markers to distinguish these populations. Moreover, there is a lack of standardization regarding the isolation methods of EV subgroups and the procedures used typically purify mixtures of vesicle types [81]. Therefore, the International Society of Extracellular Vesicles (ISEV) recently endorsed the term “extracellular vesicle” as the generic term for the isolated and studied vesicles if authors cannot establish specific markers [82]. We will use this term during the complete review, as well as in the cases where authors claimed different terms.

### 3.2. Role of EVs

#### 3.2.1. Physiological Role

EVs have a physiological role in intercellular communication, transferring proteins, lipids, and nucleic acids and thereby influencing the function of the recipient cell. The packing of this information in the vesicles provides protection of the molecules and simultaneous delivery of different messengers to remote locations [81]. EVs also have a role in immune response regulation. They act on the innate immune system as paracrine messengers and have been mainly described as pro-inflammatory mediators [83,84,85]. The functional components associated with EVs include, for example, miRNAs, fibronectin, and cytokines [86,87,88]. However, the role of EVs in innate immunity is complex [81]. EVs also play a role in the acquired immune response, in both the origin and progress, acting at different levels and on different cells [81].

It was only in 2004 that EVs in urine were first depicted as such [76]. Analysis of the RNA content from urinary EVs showed that the majority (87%) of RNA within EVs is ribosomal RNA (rRNA), whereas only 5% of the total RNA aligned to protein coding genes and splice sites. Exploration of these protein coding genes revealed that the entire genitourinary system might be mapped within EVs. Miranda et al. (2014) concluded that the majority of the non-rRNA sequences contained in the vesicles is potentially functional non-coding RNA, which play an emerging role in cell regulation [89]. Not only non-coding RNA, but also proteins affect the function of recipient cells. It has been demonstrated that certain proteins which are excreted via urinary EVs (e.g., aquaporin-2 (AQP2) and angiotensin-converting enzyme) play a role in the water balance [76,81,90]. It was also demonstrated by Hiemstra et al. (2014) that urine EVs contain viral receptors and anti-microbial proteins and peptides that could inhibit the growth of pathogenic and commensal *Escherichia coli* and induce bacterial lysis. In this way, EVs are innate immune effectors that contribute to host defense within the urinary tract [91]. 

#### 3.2.2. Role of EVs in Tumor Progression

Recent studies have shown that the crosstalk between tumor cells and the surrounding tissue plays a crucial role in cancer progression [92]. In addition to soluble molecules, EVs are involved in this process by reprogramming the tumor microenvironment and generating an invasion-promoting environment [68,69].

Tumor EVs contribute to cancer progression by influencing different immune cells. They can have an effect on anti-tumor effector T cells and prevent T-cell activation. They can also modulate other crucial components of the immune response such as myeloid and dendritic cells, impacting on the functional properties of the innate immunity [93]. Szajnik et al. (2010) also demonstrated that tumor-derived EVs induce regulatory T cells (T_reg_), promote T_reg_ expansion, upregulate their suppressor function, and enhance T_reg_ resistance to apoptosis. This interaction between tumor EVs and T_regs_ induces peripheral tolerance by tumors and supports immune evasion of human cancers [94]. Tumor EVs also seem to suppress natural killer cells and induce EV-mediated immune evasion in cancer and promote tumor growth [95,96].

Tumor EVs can also have a direct pro-tumor effect on the microenvironment. They contain protein and genetic molecules that they can transfer to distant cells. Recent evidence has shown that tetraspanins on tumor EVs are able to promote tumor growth by their capacity to induce systemic angiogenesis in tumors and tumor-free tissue [93,97]. The composition of tumor EVs can vary depending on the conditions of the secreting cells. For example, during hypoxia, tumor cells contain an increased pro-angiogenic and metastatic potential; 50% of the secreted proteins involved in this process were associated with tumor EVs [98]. Tumor EVs can also modulate stroma and the extracellular matrix that supports tumor growth, vascularization, and metastasis [99]. 

### 3.3. EV Biomarkers for Bladder Cancer

Not only the role of EVs in tumor biology but also their origin and content and the fact that they are easily accessible in body fluids render EVs a promising source of diagnostic biomarkers in oncology as well as other diseases [100,101]. Urinary EVs provide a targeted view into the urogenital tract to enhance the detection of urological diseases or tumors and their progression [101,102,103]. Researchers have also investigated the role of tumor-derived EVs in bladder cancer. Franzen et al. (2015), for example, showed that urothelial cells undergo epithelial-to-mesenchymal transition after exposure to EVs of MIBC. This process has been implicated in the initiation of metastasis for cancer progression [104]. Liang et al. (2017) demonstrated that the concentration of CD63-positive EVs in urine from patients with bladder cancer was significantly higher compared to that of healthy individuals [105]. This is also seen in other types of cancer. In addition, these reports show that urinary EVs can be a source of biomarkers for bladder cancer diagnostics. The search for EV biomarkers for bladder cancer is extensive and many potential biomarkers are described in the literature. Here, we discuss recently discovered potential urinary EV biomarkers for bladder cancer. Table 3 gives an overview of the described urinary EV-related protein and genetic biomarkers.

#### 3.3.1. Urinary EV-Related Protein Biomarkers

In addition to the detection of the concentration of EVs found in the urine, the cargo of EVs originating from bladder cancer cells can contain a specific profile [13]. Some proteomic analyses of urinary EVs from bladder cancer patients and healthy controls have already been performed [106,107,108,109,110,111,112]. Smalley et al. (2008) found that eight proteins were elevated in their isolated particles from bladder cancer patients [108]. They include five proteins associated with the epidermal growth factor receptor (EGFR) pathway, known to be deregulated in bladder cancer: Eps15 Homology (EH)-domain-containing protein 4, epidermal growth factor receptor kinase substrate 8-like protein 1 (EPS8L1), epidermal growth factor receptor kinase substrate 8-like protein 2 (EPS8L2), Guanosine-5′-triphosphate hydrolyzing enzyme NRas (GTPase NRas), and mucin 4. The last two are also seen as a biomarker for various other forms of cancer [113]. Moreover, retinoic acid-induced protein 3, resistin, and alpha subunit of Gs (G protein alpha s) GTP binding protein were upregulated. Galectin-3-binding protein was underexpressed, although normally elevated in the plasma of patients with a variety of carcinomas [108]. It is noteworthy that this study was carried out using only five healthy individuals and four bladder cancer patients, with no detectable hematuria. They used differential ultracentrifugation (UC) to isolate the EVs using a centrifugation step at 200,000 × *g* [114]. Welton et al. (2010) examined EVs isolated from the HT1376 bladder cancer cell line. They used a sucrose gradient for the isolation of the vesicles and identified 353 proteins using a liquid chromatography (LC) matrix-assisted laser desorption/ionization (MALDI) mass spectrometry (MS) workflow, based on a minimum of two identified peptides. They also used EVs isolated from the urine of three patients with transitional carcinoma of the bladder and four healthy controls. This resulted in the identification of elevated levels of CD36, CD44, 5T4, basigin, and CD73 in bladder cancer [115]. Beckham et al. (2014) found that EVs isolated from high-grade bladder cancer cell lines as well as the urine of patients with high-grade bladder cancer promoted angiogenesis and migration of bladder cancer cells and endothelial cells and thus tumor progression. This might be mediated through the delivery of EGF-like repeat and discoindin I-like domain-containing protein 3 (EDIL3), an angiogenic and cancer-associated integrin ligand that activates EGFR signaling. EVs purified from the urine of patients with high-grade bladder cancer contained significantly higher EDIL3 levels than urinary EVs from healthy individuals [109]. In addition, tumor-associated calcium-signal transducer 2 (TACSTD2) was seen as a candidate biomarker for bladder cancer [110]. It is a cell-surface glycoprotein with low to no expression in normal tissues and is overexpressed in a variety of carcinomas [116,117,118,119,120,121,122,123]. α-1-anti-trypsin and histone cluster 1 H2B family member K (H2B1K) were also identified as potential diagnostic and prognostic markers for bladder cancer [107]. Patients with detectable H2B1K had a higher risk of recurrence and progression. The expression of the markers also correlated with grading. Furthermore, periostin might be a potential biomarker [112]. Higher levels of periostin were found in urinary EVs from bladder cancer patients than those of controls. Periostin-rich EVs increase aggressiveness and promote progression, so it is expected that they are associated with a poor clinical outcome. The authors also found additional proteins, namely, Beta-hexosaminidase subunit beta (HEXB), S100A4, Staphylococcal nuclease domain-containing protein 1 (SND1), Transaldolase 1 (TALDO1), and EH domain-containing protein 4 (EHD4), that might be interesting [111].

#### 3.3.2. Urinary EV-Related Genetic Biomarkers

Another promising category of biomarkers are the different types of RNAs within EVs (e.g., lncRNAs, miRNAs, and mRNAs). A pilot study was carried out by Perez et al. (2014) and they found that four genes were differentially expressed in urinary vesicles. LAG1 longevity assurance homolog 2 (LASS2) and Polypeptide N-acetylgalactosaminyltransferase 1 (GALNT1) were present in bladder cancer patients, whereas Rho guanine nucleotide exchange factor 39 (*ARHGEF39)* and Forkhead box protein O3 (*FOXO3)* were only found in controls. However, the small number of samples and the high variability in the detected transcripts for each sample reduce the impact of this study [124]. Berrondo et al. (2016) showed that certain types of lncRNA are enriched in high-grade MIBC compared to those in healthy controls [125]. One of them is HOX transcript antisense RNA (HOTAIR), that facilitates tumor initiation and progression. Elevated levels of HOTAIR are also correlated with recurrence and poor progression [126]. Other types of known tumor-associated lncRNA that are elevated in urinary EVs include the HOXA cluster antisense RNA 2 (HOX-AS-2) and the metastasis-associated lung adenocarcinoma transcript 1 (MALAT-1) [125]. Armstrong et al. (2015) identified a number of miRNAs upregulated in urinary EVs from bladder cancer patients, for example, miR-4454, miR-720, miR-21, miR-205-5p, and miR-200c-3p [127]. Some miRNAs were upregulated in urinary EVs but not in blood plasma. Another recent study identified 26 miRNAs that were significantly dysregulated in patients with high-grade disease compared to healthy controls [128]. Of the 26 miRNAs, 23 were downregulated and 3 were upregulated. miR-375 was identified as a biomarker for high-grade bladder cancer, while miR-146a could identify low-grade disease. In the study undertaken by Baumgart et al. (2017), 15 miRNAs were identified that were significantly altered in EVs of MIBC compared to NMIBC [129]. Matsuzaki et al. (2017) also identified interesting miRNAs: 5 miRNAs were overexpressed in urinary EVs of urothelial carcinoma patients, but miR-21-5p was the most potent biomarker [130]. It was also overexpressed in urinary EVs from bladder cancer patients with negative urine cytology.

## 4. Discussion and Conclusions

Early and recurrent bladder cancer detection must still be improved due to the limitations of the current diagnostic methods of cystoscopy and cytology. Many different non-invasive urine tests are being investigated to improve the diagnostic standards currently used. This review gives an overview of the current FDA-approved tests: NMP-22 BC test, NMP-22 BladderChek®, BTA stat®, BTA TRAK®, ImmunoCyt™/uCyt+™, and UroVysion®. However, they lack sensitivity or specificity, especially for low-grade and early-stage bladder cancer tumors and recurrent diagnoses. Many false positives are detected. A possible explanation is the low abundance of certain biomarkers in an early stage of the disease and the influence of therapy on the detection of a recurrent diagnosis. This is why these tests have not replaced the current diagnostic standards of cystoscopy and cytology.

In the literature, many more potential urinary biomarkers for bladder cancer diagnostics are described that have not been implemented in an FDA-approved test. In this review, we give a non-exhaustive overview of these urinary biomarkers (Table 2). More examples can be found in other recent reviews [13,66,67]. However, some have the same limitations as the FDA-approved tests (i.e., lack sensitivity and specificity for low-grade and early-stage tumors) and further validation of these potential biomarkers is still needed.

EVs are promising as a source of biomarkers for the diagnosis and follow-up of diseases including cancer. They originate directly from tumor cells and contain an interesting cargo that is involved in tumor development. Their potential as a source of biomarkers for bladder cancer have been recently explored in several studies, which are described above. EV protein markers [44,106,107,108,109,110,111,112,115] as well as EV genetic markers [60,124,125,126,127,128,129,130] could be of great interest. 

Despite the discoveries of these potential EV biomarkers, more research is needed to prove their clinical utility. There are some limitations concerning these studies. Small and heterogenic patient populations are often used [108,128]. Variations in EV cargo exist; large samples sizes are therefore needed. Furthermore, the isolation of EVs from liquid biopsies is still not optimal and standardized due to their small and heterogeneous size range, which complicates the comparison of the results because diverse methods are used [68,131,132,133]. Some isolation methods used in these published studies result in the isolation of non-pure EV fractions. For example, centrifugation steps at 100,000 to 200,000 × *g* cause the disruption of EVs and contaminants like protein aggregates are co-pelleted [108,114,134]. This could lead to the false-positive identification of EV biomarkers. Some of the methods used have a low EV-TRACK score (evtrack.org) [23,135], indicating that extra characterization steps are needed to evaluate the quality of the EV samples.

The limitations of these studies show that there is a clear interest for easy, high-throughput, reproducible, and automatable EV recovery and analysis methods. Microfluidic chip-based technologies may revolutionize the field since lower sample volumes will be necessary and this results in more high-throughput for a clinical setting. Efforts are already been made for different types of body fluids [105,136,137,138]. Not only should EV isolation procedures be standardized, but standard procedures concerning the handling of urine should also exist. Pre-analytical variables such as urine collection, use of protease inhibitors, and storage and shipping conditions influence the composition of urine. In this way, it influences the biomarker discovery. Although the potential impact of these pre-analytical factors on EV studies is increasingly recognized, few efforts have been made to investigate and set up standards for urine [100,139]. Furthermore, for the identification of potential clinical valuable biomarkers, it is essential to use unbiased and unsupervised high-throughput discovery omics-based approaches. For example, for protein biomarker discovery using mass spectrometry approaches, it is crucial to have good quality tandem mass spectrometry data, good search criteria, and bioinformatic analysis. This will influence the confidence of the potential biomarker identification list. Biomarker panels should also be validated by negative controls, such as prostate and kidney cancer and hematuria, to determine their specificity for bladder cancer. As is mentioned in almost all published studies, all potential biomarkers still need further validation.

Studies carried out over the past years are encouraging for showing the value of EVs as a source of biomarkers. Standardized isolation protocols for EVs, correct sample sizes in biomarker discovery and validation studies, and state-of-the-art next-generation gene sequencing and mass spectrometry-based techniques must be used to produce excellent EV biomarker studies. These optimizations will eventually lead to the real potential uses of EVs in clinical settings. Characterizing the cargo of EVs from patients and controls may not only lead to the development of potential biomarkers, but also generate insights into tumor biology in general. 

## Figures and Tables

**Table 1 ijms-20-00821-t001:** Commercially available FDA-approved tests for bladder cancer. The biomarker and assay type of the tests are included in the table. The mean and range (between brackets) of the overall sensitivity and specificity are also shown. Adapted from [12,14,23,24,29].

Test	Biomarker	Assay Type	Sensitivity (%)	Specificity (%)
NMP22® BC test	NMP-22	Sandwich immunoassay	69 (26–100)	77 (41–92)
NMP22® BladderChek®	NMP-22	Sandwich immunoassay	58 (51–85)	88 (77–96)
BTA stat®	Complement factor H-related protein	Colorimetric immunoassay	64 (29–83)	77 (56–86)
BTA TRAK®	Complement factor H-related protein	Sandwich immunoassay	65 (53–91)	74 (28–83)
ImmunoCyt™	Carcinoembryonic antigen and 2 mucins	Immunofluorescence cytology	78 (52–100)	78 (63–79)
UroVysion™	Aneuploidy of chromosomes 3, 7, 17 and loss of 9p21 locus	Multitarget FISH	63 (30–86)	87 (63–95)

**Table 2 ijms-20-00821-t002:** Non-exhaustive overview of non-FDA approved urine biomarkers for bladder cancer. Twist-related protein 1 (TWIST1), Protein odd-skipped-related 1 (OSR1), Single-minded homolog 2 (SIM2), Homeobox protein OTX1 (OTX1), Homeobox protein Meis1 (MEIS1), One cut domain family member 2 (ONECUT2)

Biomarkers	References
Urine protein biomarkers	
BLCA-1 and BLCA-4	[30,31]
Hyaluronidase	[33,34]
Cytokeratins 8, 18, 19	[35,37]
Survivin	[40,41,42]
ProEGF, SAA4, APOA1, APOA2, APOB, APOC2, APOC3, APOE	[44]
CCL18, PAI-1, CD44	[46]
Urine metabolite biomarkers	
Lactate, β-hydroxypyruvate, palmitoyl sphingomyelin, phosphocholine, arachidonate, BCAAs, adenosine, succinate	[49]
Urine genetic biomarkers	
FGFR3	[50,51,52,65]
p53	[65]
CDK1, HOXA13, MDK, IGFBP5	[53]
PON2	[54]
miR-26a, miR-93, miR-191, miR-940	[59]
Methylation of TWIST1, OSR1, SIM2, OTX1, MEIS1, ONECUT2	[63]

**Table 3 ijms-20-00821-t003:** Non-exhaustive overview of urinary EV biomarkers for bladder cancer. The EV isolation method used in the study is also shown.

Biomarkers	EV Isolation Method	References
Urine EV protein biomarkers		
EH-domain-containing protein 4, EPS8L1, EPS8L2, GTPase NRas, Mucin 4, retinoic acid-induced protein 3, resistin, alpha subunit of GsGTP binding protein	Differential UC	[108]
CD36, CD44, 5T4, basigin, CD73	Differential UC + sucrose cushion	[115]
EDIL3	Differential UC + filtration + sucrose cushion	[109]
TACSTD2	Differential UC	[110]
α-1-anti-trypsin, H2B1K	Differential UC	[107]
Periostin	Differential UC	[112]
HEXB, S100A4, SND1, TALDO1, and EHD4	Differential UC	[111]
Urine EV genetic biomarkers		
LASS2, GALNT1, ARHGEF39, FOXO3	Filtration + Differential UC	[124]
HOTAIR, HOX-AS-2, MALAT-1	Differential UC	[125]
miR-4454, miR-720, miR-21, miR-205-5p, miR-200c-3p	Differential UC + Urine Exosome RNA Isolation Kit (Norgen)	[127]
miR200c, miR93, miR940, miRlet7b, miR191, miR21, miR15a	Differential UC	[60]
miR-375 (high-grade), miR-146a (low-grade)	Filtration + Differential UC	[128]
miR-200-3p	Differential centrifugation + Total Exosome Isolation Kit (Life Technologies)	[129]
miR-21-5p	Differential UC	[130]

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
