# Peer review of "Bladder Cancer Diagnosis and Follow-Up: The Current Status and Possible Role of Extracellular Vesicles"

_ijms, 2019, doi:10.3390/ijms20040821_

Reviewer 1 Report

In general, it is important to publish reviews like "The possible role of extracellular vesicles for bladder cancer diagnosis and follow-up".

In most cases, especially for mRNA and also for urinary based fast Tests the literature is too old.

Regarding cytokeratins there should be a much better review of the literature. E.g. the Information About UBC rapid is based on a study (1999 and 2001) with a different test for this marker.

Also tables for a better overview should be added.

Author Response

Dear reviewer,

Thank you for the opportunity to revise our manuscript. Please find attached a point-by-point response to your comments. It is our belief that the review is substantially improved after making the suggested edits of all reviewers. We look forward to hearing from you.

Sincerely,

All authors

In general, it is important to publish reviews like "The possible role of extracellular vesicles for bladder cancer diagnosis and follow-up".

In most cases, especially for mRNA and also for urinary based fast Tests the literature is too old.

We added some recent references for the urinaryebased FDA approved tests. Also a recent study of Bacchetti et al. 2018 was added about paraoxonase-2 as example of an mRNA biomarker for bladder cancer, besides the well-known Cxbladder® test that was described. We also refer now to the recent review of Wieczorek et al. 2018 for a more recent overview on mRNA, microRNA and lncRNA bladder cancer markers.

Regarding cytokeratins there should be a much better review of the literature. E.g. the Information About UBC rapid is based on a study (1999 and 2001) with a different test for this marker.

We kept the references of Sanchez-Carbayo et al. (1999/2001) but also added a more recent research article from Ecke et al. 2017 about the UBC-rapid test. For the CYFRA 21-1 test, we added the more recent references of Jeong et al. 2012 and Guo et al. 2016.

Also tables for a better overview should be added.

There are 3 tables in the paper that summarize the review. We adapted some tables and added some columns with extra information to give a better overview.

Reviewer 2 Report

In the manuscript “The possible role of extracellular vesicles for bladder cancer diagnosis and follow-up”, the authors review diagnostic tools for bladder cancer diagnosis and follow-up. Obviously, there is a great demand for minimally invasive diagnostic tool and urine is a promising source for relevant biomarkers for disease in the urogenital tract. In the last section the authors focus on the potential of urinary extracellular vesicles (EVs) as source of biomarkers. A review on this topic is certainly timely and of interest.

 Major comments:

-          The title “The possible role of extracellular vesicles for bladder cancer diagnosis and follow-up” strongly focuses on EVs. However, only a small part (2 of the 9 pages) of the review focuses on EVs and their role in diagnosis. Therefore the title is not well balanced and a more title covering bladder cancer diagnostics in general would be more fitting.

-          Alternatively, (urinary) EVs in bladder cancer diagnosis and follow-up should be discussed in far more detail.  The current manuscript (only) summarizes the EV related protein and RNA biomarkers and only shortly discusses the use of EVs in diagnostic analysis. The manuscript would gain impact and relevance when a in depth discussion on the potentials, limitations and challenges of EV for diagnosis. A clear distinction between the use of urinary EVs and blood derived EVs would be of value.

-          The authors state in several parts of the review that they present a non-exhaustive summary or list. It would strengthen the manuscript if these summaries and lists would be complete. If that is not feasible, the authors should indicate the parameters used for the selection of markers and assays.

Minor comments:

-          Page 2. The working principles of the FDA approved urine tests should be discussed in more detail.

-          Page 5. The section 3 title is “Exosomes and Extracellular Vesicles”. However, exosomes is an extracellular vesicle sub-type. The authors should adapt the section title.

-          Page 5/6. In this review on bladder cancer diagnostics, the discussion of the role of EVs in normal physiology and cancer is as long as the summary/discussion on EVs in bladder cancer diagnosis. The authors should more balance these discussions.  

-          Page 8. Despite of the importance of table 3 in this review, the text does not refer to this table. The authors should include the reference to table 3.

Author Response

Dear reviewer,

Thank you for the opportunity to revise our manuscript. We appreciate the constructive comments. Please find attached our point-by-point response to your comments. It is our belief that the manuscript is substantially improved after making the suggested edits from all reviewers.

Sincerely,

All authors

In the manuscript “The possible role of extracellular vesicles for bladder cancer diagnosis and follow-up”, the authors review diagnostic tools for bladder cancer diagnosis and follow-up. Obviously, there is a great demand for minimally invasive diagnostic tool and urine is a promising source for relevant biomarkers for disease in the urogenital tract. In the last section the authors focus on the potential of urinary extracellular vesicles (EVs) as source of biomarkers. A review on this topic is certainly timely and of interest.

Major comments:

-          The title “The possible role of extracellular vesicles for bladder cancer diagnosis and follow-up” strongly focuses on EVs. However, only a small part (2 of the 9 pages) of the review focuses on EVs and their role in diagnosis. Therefore the title is not well balanced and a more title covering bladder cancer diagnostics in general would be more fitting.

We adapted the title in “B+ladder cancer diagnosis and follow-up: the current status and possible role of extracellular vesicles”. This title covers indeed better the content of the review.

-          Alternatively, (urinary) EVs in bladder cancer diagnosis and follow-up should be discussed in far more detail.  The current manuscript (only) summarizes the EV related protein and RNA biomarkers and only shortly discusses the use of EVs in diagnostic analysis. The manuscript would gain impact and relevance when a in depth discussion on the potentials, limitations and challenges of EV for diagnosis. A clear distinction between the use of urinary EVs and blood derived EVs would be of value.

Since we focus on bladder cancer diagnosis, we’d like to describe the value of urinary EVs in bladder cancer diagnosis since urine is in direct contact of the bladder tumor and is a more easily accessible body fluids than blood. We adapted the title that fits the content of the review more. In the discussion, we discuss the potentials, limitations and challenges of EV for diagnosis. We made some adaptions in the discussion that give the discussion more structure. We hope these adaptions improved the manuscript.

-          The authors state in several parts of the review that they present a non-exhaustive summary or list. It would strengthen the manuscript if these summaries and lists would be complete. If that is not feasible, the authors should indicate the parameters used for the selection of markers and assays.

For the non-FDA approved urine biomarkers and EV biomarkers, we give a non-exhaustive overview since literature is extensive. For each type of biomarker, we try to give a few examples for bladder cancer diagnostics. In the revised version of the manuscript, we tried to make this clear. We also refer to more extensive reviews in the text now, focusing for example specifically on a certain type of biomarker for bladder cancer diagnostics such as Santoni et al. 2018 and Wieczorek et al. 2018.

  Minor comments:

-          Page 2. The working principles of the FDA approved urine tests should be discussed in more detail.

Since the most interesting section of the review is about the extracellular vesicles in bladder cancer diagnostics, we do not like to expand the section about the FDA approved test with more details. However, indeed the working principle of some FDA approved test were missing. We made some minor edits in the text to make the working principle clear and also added 2 columns in Table 1 that explain the assay type and biomarker detected for each FDA approved test.

 -          Page 5. The section 3 title is “Exosomes and Extracellular Vesicles”. However, exosomes is an extracellular vesicle sub-type. The authors should adapt the section title.

We changed the title in “Extracellular vesicles” since they cover indeed already the most interesting subtype, exosomes. In section 3.1 we explain why we use the term “extracellular vesicle” instead of “exosome” according the MISEV guidelines.

 -          Page 5/6. In this review on bladder cancer diagnostics, the discussion of the role of EVs in normal physiology and cancer is as long as the summary/discussion on EVs in bladder cancer diagnosis. The authors should more balance these discussions.  

We made some adapations in section “3. Extracellular vesicles” that hopefully balance the review more. Please see the revised manuscript.

-          Page 8. Despite of the importance of table 3 in this review, the text does not refer to this table. The authors should include the reference to table 3

We now refer to table 3 in the beginning of section “3.3 EV biomarkers for bladder cancer”.

Reviewer 3 Report

In the proposed review the authors aimed to give an overview of both approved and innovative urine tests to detect bladder cancer with an interesting section concerning the use of extracellular vesicles from urinary as cancer biomarkers.

Overall the topic is interesting and most of the proposed bibliography is quite recent.

In order to make the manuscript more incisive and easy to read I propose the following modification in the review’s architecture.

The “non-exhaustive” overview of FDA approved test is redundant and distracting from the main topic indicated in the title. I suggest substituting section 2.1 with an enriched table1, in which a further column indicating the limits of the approved could be added.

·         In line 124 please add references concerning the use of APO as bladder cancer biomarkers.

·         Please describe further the use of the indicated “metabolites” as biomarkers (line 129)

·         The sentence in line 134 starting with Epigenetic factors……is repeated in line 149.

·         In the same section2.2, it is not clear the correlation among coding and non-coding RNAs, epigenetic and FGFR mutation.

Finally, in my opinion, the use of short conclusions at the ending of each section penalizes the fluidity of the manuscript.

Author Response

Dear reviewer,

Thank you for the opportunity to revise our manuscript. It is our belief that the manuscript is substantially improved after making the suggested edits of the reviewers. Please find attached our point-by-point response to your comments.

Sincerely,

All authors

In the proposed review the authors aimed to give an overview of both approved and innovative urine tests to detect bladder cancer with an interesting section concerning the use of extracellular vesicles from urinary as cancer biomarkers. Overall the topic is interesting and most of the proposed bibliography is quite recent.

In order to make the manuscript more incisive and easy to read I propose the following modification in the review’s architecture.

The “non-exhaustive” overview of FDA approved test is redundant and distracting from the main topic indicated in the title. I suggest substituting section 2.1 with an enriched table1, in which a further column indicating the limits of the approved could be added.

We decided to keep section 2.1 since other reviewers even wanted us to extend this section with more details on the working principles of these tests. However, we changed the title of the review that better cover the content of the review and also made some edits in table 1 and section 2.1 (Please see revised manuscript).

·         In line 124 please add references concerning the use of APO as bladder cancer biomarkers.

This part in section 2.2 was also adapted. We added the reference of Kumar et al. 2015 (see text, now lines 146-150)

·         Please describe further the use of the indicated “metabolites” as biomarkers (line 129)

We adapted this part of the manuscript, see lines 155-160 in the revised version.

·         The sentence in line 134 starting with Epigenetic factors……is repeated in line 149.

We deleted this sentence in line 134 and made a separate paragraph on epigenetic factors (see text lines 187-193).

·         In the same section2.2, it is not clear the correlation among coding and non-coding RNAs, epigenetic and FGFR mutation.

We changed the structure of section 2.2. Please see text and hopefully this section is more clear now.

Finally, in my opinion, the use of short conclusions at the ending of each section penalizes the fluidity of the manuscript

We tried to include the ‘short conclusion paragraphs’ of each section in the section itself as paragraph which makes the text more readable (see revised version of the review).

Round  2

Reviewer 1 Report

better

Reviewer 2 Report

In the manuscript “The possible role of extracellular vesicles for bladder cancer diagnosis and follow-up”, the authors review diagnostic tools for bladder cancer diagnosis and follow-up. Obviously, there is a great demand for minimally invasive diagnostic tool and urine is a promising source for relevant biomarkers for disease in the urogenital tract. In the last section the authors focus on the potential of urinary extracellular vesicles (EVs) as source of biomarkers. A review on this topic is certainly timely and of interest.

All comments and suggestions have been addressed in the revised manuscript.